# The Impact of Amino Acids on Postprandial Glucose and Insulin Kinetics in Humans: A Quantitative Overview

**DOI:** 10.3390/nu12103211

**Published:** 2020-10-21

**Authors:** Bart van Sloun, Gijs H. Goossens, Balazs Erdos, Michael Lenz, Natal van Riel, Ilja C. W. Arts

**Affiliations:** 1TiFN, 6700 AN Wageningen, The Netherlands; g.goossens@maastrichtuniversity.nl (G.H.G.); balazs.erdos@maastrichtuniversity.nl (B.E.); ilja.arts@maastrichtuniversity.nl (I.C.W.A.); 2Maastricht Centre for Systems Biology (MaCSBio), Maastricht University, 6200 MD Maastricht, The Netherlands; 3Department of Human Biology, NUTRIM School of Nutrition and Translational Research in Metabolism, Maastricht University, 6200 MD Maastricht, The Netherlands; 4Institute of Organismic and Molecular Evolution, Johannes Gutenberg University Mainz, D 55099 Mainz, Germany; michael.l.lenz@googlemail.com; 5Preventive Cardiology and Preventive Medicine-Centre for Cardiology, University Medical Center of the Johannes Gutenberg University Mainz, D 55131 Mainz, Germany; 6Department of Biomedical Engineering, Eindhoven University of Technology, 5600 MB Eindhoven, The Netherlands; N.A.W.v.Riel@tue.nl; 7Department of Epidemiology, CARIM School for Cardiovascular Diseases, Maastricht University, 6200 MD Maastricht, The Netherlands

**Keywords:** systematic review, amino acids, kinetics, dynamics, glucose homeostasis, insulin secretion, insulin sensitivity, obesity, type 2 diabetes, time series data

## Abstract

Different amino acids (AAs) may exert distinct effects on postprandial glucose and insulin concentrations. A quantitative comparison of the effects of AAs on glucose and insulin kinetics in humans is currently lacking. PubMed was queried to identify intervention studies reporting glucose and insulin concentrations after acute ingestion and/or intravenous infusion of AAs in healthy adults and those living with obesity and/or type 2 diabetes (T2DM). The systematic literature search identified 55 studies that examined the effects of l-leucine, l-isoleucine, l-alanine, l-glutamine, l-arginine, l-lysine, glycine, l-proline, l-phenylalanine, l-glutamate, branched-chain AAs (i.e., l-leucine, l-isoleucine, and l-valine), and multiple individual l-AAs on glucose and insulin concentrations. Oral ingestion of most individual AAs induced an insulin response, but did not alter glucose concentrations in healthy participants. Specific AAs (i.e., leucine and isoleucine) co-ingested with glucose exerted a synergistic effect on the postprandial insulin response and attenuated the glucose response compared to glucose intake alone in healthy participants. Oral AA ingestion as well as intravenous AA infusion was able to stimulate an insulin response and decrease glucose concentrations in T2DM and obese individuals. The extracted information is publicly available and can serve multiple purposes such as computational modeling.

## 1. Introduction

Glucose is a key substrate for many different types of cells and tissues, and as such plays an important role in human metabolism [1]. Blood glucose concentrations are tightly regulated to prevent hypoglycemia and hyperglycemia, thereby ensuring normal body functions. Glucagon and adrenalin are the key hormones responsible for elevating blood glucose concentration (for example during fasting or exercise), whereas insulin lowers blood glucose concentrations (for example following meal intake) [2]. This complex regulatory system involves various organs, including the gut, pancreas, liver, adipose tissue and skeletal muscle. Impairment of glucose homeostasis increases the risk of developing chronic cardiometabolic diseases such as type 2 diabetes mellitus (T2DM) and cardiovascular disease, highlighting the importance of adequate control of blood glucose concentrations [3].

Amino acids (AAs) are involved in the regulation of insulin secretion through their effects on β-cells, causing a rise in the ATP/ADP ratio, suppression of ATP-sensitive potassium channels and activation of voltage-gated Ca^2+^ channels [4]. The resulting calcium influx allows exocytosis of insulin granules from the β-cells. The insulinotropic effect of AA administration in humans was studied in the 1960s by Floyd et al. [5,6]. An intravenous infusion of an AA mixture, consisting of essential l-AA, increased plasma insulin concentration and subsequently lowered blood glucose concentrations in healthy people [5]. A similar response was also demonstrated following infusion of single AAs [5,6]. Interestingly, however, there seemed to be large differences in the capacity of the various AAs to stimulate insulin secretion [5]. A synergistic effect of simultaneous glucose and AA ingestion was found in several studies, where co-ingestion of AAs with glucose increased insulin secretion more than the sum of the individual effects [6,7,8,9]. The insulinotropic effect of AA administration has also been demonstrated in patients with T2DM. For example, co-ingestion of a protein hydrolysate/AA mixture with carbohydrates induced a more pronounced increase in plasma insulin concentrations compared to intake of carbohydrates alone, not only in healthy people but also in patients with T2DM [9]. Of note, the metabolic phenotype may influence the magnitude of the glucose and insulin responses following ingestion of AAs.

Despite decades of research, there is no quantitative overview available that describes the effects of individual AAs on glucose and insulin kinetics in individuals with different metabolic phenotypes. These data are needed as the type and amount of AA intake, the administration route (i.e., oral versus intravenous) and study population seems to affect the insulin and glucose responses to AAs. We hypothesize that AAs exert distinct effects on glucose and insulin dynamics, which are further influenced by the metabolic phenotype as well as route of administration. A better mechanistic understanding would therefore allow for more targeted nutrition studies. This information could also aid in the improvement of physiology-based computational models of the glucose regulatory system, which have been developed and approved for pre-clinical research [10]. Therefore, we performed a systematic literature search and extracted the original data, which we made publicly available, to obtain better insight into the quantitative acute effects of individual AAs on postprandial glucose and insulin dynamics in humans, taking the metabolic phenotype into account.

## 2. Materials and Methods

Details of the systematic review were registered in the PROSPERO International Prospective Register of Systematic Reviews (registration number CRD42020155067).

### 2.1. Search Strategy

Studies assessing the quantitative acute effects of AAs on postprandial plasma glucose and insulin concentrations were retrieved from the PubMed database between February 2018 and February 2020. The search strategy contained multiple (combinations of) main keywords appropriate for the topic of interest (“Amino acids” AND “Postprandial” AND (“Glucose” OR “Insulin”)). The detailed search strategy is provided in the Appendix A. Searches were not limited by article publication date. From the identified articles, the titles and abstract were assessed first, and if considered relevant for the present systematic review, the full text of the article was examined in detail.

### 2.2. Selection Criteria

Criteria for study inclusion were set according to the Population-Intervention-Comparator-Outcomes-Study design (PICOS) format (Table 1). Eligible studies included healthy adults as well as people living with overweight/obesity and T2DM. Only acute studies that evaluated the effects of oral AA ingestion and/or intravenous AA administration on glucose and insulin concentrations were included. Studies had to be original research and be written in English. Labels were added to the eligible articles, describing the subject characteristics, type of AA and route of AA administration.

### 2.3. Data Extraction

Data were extracted from eligible studies and entered into an Excel (2016, Microsoft Corporation) spreadsheet. The following items were extracted: bibliographic details (title; authors; year; journal), study population (health status; number of subjects; sex; age; body mass index (BMI); weight), intervention (type; dose; method of administration; duration), study design and the outcome measurements of interest (glucose concentrations; insulin concentrations). In the absence of exact numerical values for the outcome measurements of interest in the original articles, figures were digitized using graph digitizing software (Graph Grabber version 2.0, Quintessa). After loading the figures in the digitizing software, the *x*- and *y*-axis was set to correctly map the image pixels to the corresponding data values in the figure. Data points (i.e., means and their corresponding standard errors or standard deviations) were determined by manually clicking them and were subsequently stored in Excel data sheets for further processing. The extracted data from the 55 studies is publicly available (https://doi.org/10.34894/RNZI0A) and can serve multiple purposes such as computational modeling.

### 2.4. Data Processing

The extracted data was imported into MATLAB [11]. Measurement units were converted to mg/dL for glucose and µU/mL for insulin. The iAUC was calculated according to the trapezoidal rule in MATLAB, and normalized to one minute by dividing with the time span of the response. The time to glucose and insulin peaks and their concentrations were identified from the extracted data by selecting the data-points with the highest glucose and insulin concentrations, and their corresponding time points in MATLAB. No statistical analysis was performed. The mention of significance in this review refers to the statistical analysis performed in the original studies.

## 3. Results

### 3.1. Literature Search

The PubMed search identified 10,311 unique records (Figure 1). Titles and/or abstracts were screened, which resulted in the exclusion of 9678 studies. Three additional records were identified through searching reference lists. The remaining 636 full-text articles were retrieved and assessed for eligibility based on the selection criteria (Table 1). From these full-text articles, a total of 548 articles were excluded. Reasons for exclusion were related to the target population, intervention, time-scale, outcome parameters, and article language. An additional 33 articles were excluded due to accessibility restrictions and could not be retrieved via the university library. Thus, a total of 55 studies were included in the analysis. In the studies, the effects of l-leucine (*n* = 6), l-isoleucine (*n* = 1), l-alanine (*n* = 6), l-glutamine (*n* = 1), l-arginine (*n* = 28), l-lysine (*n* = 1), glycine (*n* = 2), l-proline (*n* = 1), l-phenylalanine (*n* = 1), l-glutamate (*n* = 3), branched-chain AAs (BCAAs) (i.e., l-leucine, l-isoleucine, and l-valine) (*n* = 4), and multiple l-AAs ingested separately (i.e., l-leucine, l-arginine, l-lysine, and l-phenylalanine) (*n* = 1) on postprandial glucose and insulin concentrations were determined. When mentioning different AAs throughout the present manuscript, with the exception of glycine that has no enantiomers because it has two hydrogen atoms attached to the central carbon atom, we refer to the l-isoforms of the respective AA.

### 3.2. Leucine

Leucine study details are provided in Table 2 (section A). The time series data and the calculated kinetic parameters are visualized in Figure 2. The effect of oral leucine intake (Figure 2A) has only been examined in healthy individuals [8,12,13]. Two out of the three studies showed increased insulin concentrations (iAUC range, 0.85 to 0.95 µU/mL/min) from baseline [12] and compared to a water control group (0.28 µU/mL/min) [8]. Glucose concentrations were unchanged compared to the water control group. The study with the lowest oral leucine dose showed decreased insulin (iAUC, −1.22 µU/mL/min) [13]. Here, the insulin reached a peak (5.35 µU/mL at 15 min), followed by a decrease below baseline values.

Kalogeropoulou et al. [8] demonstrated that co-ingestion of leucine+glucose (Figure 2B) increased the insulin concentration (iAUC, 21.25 µU/mL/min) more than the sum of their individual effects (iAUC, 12.87 and 0.95 µU/mL/min for glucose and leucine ingestion, respectively), and attenuated the glucose response. The insulin concentrations increased more rapidly and reached a higher insulin peak (53.1 µU/mL) after co-ingestion of leucine+glucose compared to glucose ingestion alone (31.0 µU/mL).

The effect of intravenous leucine infusion (Figure 2C) has also been examined in healthy people only [14,15,16,17]. Three out of the four studies showed increased insulin (iAUC range, 4.47 to 8.90 µU/mL/min) [14,15,17], and decreased glucose concentrations (iAUC range, −2.52 to −6.31 mg/dL/min) compared to baseline. The study with the lowest intravenous leucine dose did not show increased insulin concentrations (iAUC, −0.45 µU/mL/min) [16]. To our knowledge, no kinetic data are available concerning leucine intake in people with obesity and/or T2DM.

### 3.3. Isoleucine

Isoleucine study details are provided in Table 2 (section B). The time series data and the calculated kinetic parameters are visualized in Appendix A. Nuttall et al. [18] demonstrated that oral isoleucine ingestion alone (Appendix A) had no significant effect on insulin concentrations, but decreased glucose (iAUC, −4.15 mg/dL/min) compared to ingestion of water (iAUC, 0.30 mg/dL/min) in healthy individuals. Co-ingestion of isoleucine+glucose (Appendix A) increased the insulin concentrations (iAUC, 21.38 µU/mL/min) more than the sum of their individual effects (iAUC, 19.0 and 0.00 µU/mL/min for glucose, and isoleucine ingestion respectively), and attenuated the glucose-stimulated glucose response in healthy individuals [18]. To our knowledge, no kinetic data are available concerning intravenous isoleucine infusion, nor for isoleucine intake in people with obesity and/or T2DM.

### 3.4. Alanine

Alanine study details are provided in Table 2 (section C). The time series data and the calculated kinetic parameters are visualized in Appendix A. The effect of oral alanine ingestion has been examined in healthy individuals [19,20,21,22], T2DM patients [22], and obese individuals [23] (Appendix A). All four studies showed increased insulin concentrations (iAUC range, 1.01 to 10.53 µU/mL/min) from baseline following oral ingestion of alanine in healthy individuals. Alanine was found to lower glucose concentrations (iAUC, −3.33 mg/dL/min) in one study [21] that had the highest alanine dose. Genuth et al. [22] demonstrated that the effect of oral ingestion of alanine on insulin is dose-dependent, with high dosing (33.8 g) leading to a larger postprandial insulin response (iAUC, 10.53 µU/mL/min) compared to low dosing (6.8 g) (iAUC, 1.01 µU/mL/min). Plasma glucose concentrations were unchanged in both interventions, with the exception of a slight but significant decrease in glucose concentrations 240 min after oral ingestion for the low alanine dose. The initial insulin response to oral alanine ingestion was similar in healthy individuals and T2DM patients, but the insulin concentrations remained elevated over a prolonged period of time in the latter [22]. Glucose concentrations were decreased (iAUC range, −12.99 to −12.29 mg/dL/min) from baseline in T2DM patients. Oral ingestion of alanine also increased insulin (iAUC, 42.67 µU/mL/min), and decreased glucose concentrations (iAUC, −10.49 mg/dL/min) from baseline in people with obesity [23]. To our knowledge, no kinetic data are available concerning co-ingestion of alanine with glucose in individuals.

Additionally, the effect of intravenous alanine infusion was examined in healthy [19,24] and obese individuals [24] (Appendix A). Rossini et al. [19] showed that, in contrast to oral alanine ingestion, intravenous alanine infusion did not alter insulin and glucose concentrations from baseline. Asano et al. [24] showed increased insulin concentrations in both obese participants (iAUC, 12.02 µU/mL/min) and healthy individuals (iAUC, 3.42 µU/mL/min). Glucose concentrations were also increased, despite not being different between groups.

### 3.5. Glutamine

Glutamine study details are provided in Table 2 (section D). The time series data and the calculated kinetic parameters are visualized in Appendix A. Greenfield et al. [25] demonstrated that oral ingestion of glutamine increased insulin concentrations compared to ingestion of water only in healthy, T2DM and obese individuals (Appendix A). The effects found were most pronounced in T2DM individuals (iAUC, 13.35 and −0.23 µU/mL/min for glutamine, and water ingestion, respectively), intermediate in obese individuals (iAUC, 6.16 and −2.73 µU/mL/min for glutamine, and water ingestion, respectively), and modest in healthy individuals (iAUC, 1.62 and −1.06 µU/mL/min for glutamine, and water ingestion, respectively). The glucose concentrations were comparable to water ingestion in these groups. To our knowledge, no kinetic data are available concerning individuals co-ingesting glutamine with glucose, nor for intravenous glutamine infusion.

### 3.6. Arginine

Arginine study details are provided in Table 2 (section E). The time series data and the calculated kinetic parameters are visualized in Appendix A. The effects of oral arginine ingestion have been investigated in healthy individuals only [26,27] (Supplemental Appendix A). One out of the two studies [27], with the largest oral arginine dose, showed increased insulin concentrations (iAUC, 1.41 µU/mL/min) compared to water intake (iAUC, 0.06 µU/mL/min), with no significant effect on glucose concentrations.

Co-ingestion of arginine+glucose (Appendix A) resulted in a similar iAUC for insulin compared to glucose ingestion alone and attenuated the glucose-stimulated glucose response [26]. Despite the lower insulin peak, the insulin concentrations remained elevated for a longer time after co-ingestion of arginine+glucose. Tang et al. [27] showed a non-significant increase in insulin concentrations when glucose was co-ingested with arginine (iAUC, 28.62 µU/mL/min) compared to glucose ingestion alone (iAUC, 19.05 µU/mL/min). Glucose concentrations were unchanged.

The effect of intravenous arginine infusion has been examined in healthy individuals [17,28,29,30,31,32,33,34,35,36,37,38,39,40,41,42,43,44,45,46], T2DM patients [42,43,44,45,47,48,49,50], and obese individuals [46,51,52,53] (Appendix A). All twenty studies [17,28,29,30,31,32,33,34,35,36,37,38,39,40,41,42,43,44,45,46] on intravenous arginine infusion in healthy individuals showed increased insulin concentrations (iAUC range, 1.58 to 45.75 µU/mL/min) from baseline. Glucose concentrations increased (iAUC range, 1.79 to 18.65 mg/dL/min) from baseline in fourteen studies [17,28,29,31,33,34,35,38,39,40,41,42,43,45]. Four additional studies showed an increase in glucose concentrations in the beginning of the study followed by a drop below baseline, resulting in a negative iAUC [36,37,44,46]. Two studies [30,32] included an intravenous saline infusion control group and showed increased insulin concentrations after intravenous arginine infusion (iAUC, 11.42 and 25.98 µU/mL/min, respectively) compared to intravenous saline infusion (iAUC, −0.64 and 5.33 µU/mL/min, respectively). Broglio et al. [30] also showed increased glucose concentrations (iAUC, 4.45 mg/dL/min) after intravenous arginine infusion compared to intravenous saline infusion (iAUC, −1.64 mg/dL/min); however, no dynamic data were provided for both insulin and glucose. Dela et al. [31] found that intravenous arginine infusion increased insulin and glucose concentrations from baseline in trained and untrained, healthy individuals, with lower insulin concentrations in trained males (iAUC, 18.40 µU/mL/min and 45.75 µU/mL/min, for trained and untrained individuals respectively). The glucose concentrations did not differ between the two groups.

All eight studies [42,43,44,45,47,48,49,50] investigating intravenous arginine infusion in T2DM individuals showed increased insulin (iAUC range, 4.09 to 21.66 µU/mL/min) and glucose concentrations (iAUC range, 5.00 to 29.75 mg/dL/min). Three studies [43,44,45] demonstrated that insulin responses were lower in T2DM patients (iAUC range, 4.09 to 13.17 µU/mL/min) compared to healthy individuals (iAUC range, 21.84 to 31.25 µU/mL/min) during intravenous arginine infusion. Efendić et al. [42] demonstrated a slightly lower insulin response in T2DM patients (iAUC, 21.66 µU/mL/min) compared to healthy individuals (iAUC, 22.49 µU/mL/min) who received intravenous arginine infusion, but the insulin concentrations remained elevated over a prolonged period of time in T2DM patients. Glucose concentrations were increased from baseline in both healthy individuals (iAUC, 5.04 mg/dL/min) and T2DM patients (iAUC, 8.44 mg/dL/min). All four studies [46,51,52,53] on intravenous arginine infusion in obese individuals showed increased insulin (iAUC range, 6.10 to 22.07 µU/mL/min) and glucose concentrations (iAUC range, 1.37 to 9.08 mg/dL/min). Maccario et al. [46] demonstrated that the arginine induced insulin response was higher in obese participants (iAUC, 22.07 µU/mL/min) than in healthy individuals (iAUC, 8.30 µU/mL/min). Glucose concentrations did not decrease below baseline values in obese participants (iAUC, 1.60 mg/dL/min) compared to healthy individuals (iAUC, −3.20 mg/dL/min).

### 3.7. Lysine

Lysine study details are provided in Table 2 (section F). The time series data and the calculated kinetic parameters are visualized in Appendix A. The effects of oral lysine ingestion [54] (Appendix A) and intravenous lysine infusion [17] (Appendix A) were examined in healthy individuals. Kalogeropoulou et al. [54] demonstrated that oral ingestion of lysine increased insulin (iAUC, 0.67 µU/mL/min) and decreased glucose concentrations (iAUC, −1.73 mg/dL/min) compared to individuals that ingested water (iAUC, −0.62 µU/mL/min, 0.35 mg/dL/min). Co-ingestion of lysine+glucose (Appendix A) resulted in a similar iAUC for insulin compared to glucose ingestion alone and attenuated the glucose-stimulated glucose response. However, insulin concentrations increased more rapidly and reached higher peak insulin concentrations (38.7 µU/mL) compared to glucose ingestion alone (35.4 µU/mL).

Intravenous lysine infusion (Appendix A) increased insulin (iAUC, 8.83 µU/mL/min) and decreased glucose concentrations (iAUC, −2.23 mg/dL/min) from baseline in healthy individuals [17]. To our knowledge, no kinetic data are available concerning lysine intake in people with obesity and/or T2DM.

### 3.8. Glycine

Glycine study details are provided in Table 2 (section G). The time series data and the calculated kinetic parameters are visualized in Appendix A. The effects of oral glycine ingestion have been examined in healthy individuals [55,56] (Appendix A). One out of the two studies showed increased insulin concentrations (iAUC, 2.29 µU/mL/min) compared to water (iAUC, −1.66 µU/mL/min), with no change in glucose concentrations [55]. Co-ingestion of glycine+glucose (Appendix A) resulted in a similar iAUC for insulin compared to glucose ingestion alone and attenuated the glucose-stimulated glucose response. However, insulin concentrations increased more slowly and reached lower peak insulin concentrations (76.7 µU/mL) compared to glucose ingestion alone (92.4 µU/mL). To our knowledge, no kinetic data are available concerning intravenous glycine infusion, nor for glycine intake in people with obesity and/or T2DM patients.

### 3.9. Proline

Proline study details are provided in Table 2 (section H). The time series data and the calculated kinetic parameters are visualized in Appendix A. Nuttall et al. [57] demonstrated that oral ingestion of proline (Appendix A) increased insulin (iAUC, 0.46 µU/mL/min) compared to intake of water (iAUC, −0.99 µU/mL/min), with no change in glucose concentrations in healthy individuals. Co-ingestion of proline+glucose (Appendix A) resulted in a comparable iAUC for insulin compared to glucose ingestion alone, and attenuated the glucose-stimulated glucose response. However, insulin concentrations increased more rapidly and reached higher peak insulin concentrations (41.2 µU/mL) compared to glucose ingestion alone (33.0 µU/mL) in healthy individuals. To our knowledge, no kinetic data are available concerning intravenous proline infusion, nor for proline intake in people with obesity and/or T2DM patients.

### 3.10. Phenylalanine

Phenylalanine study details are provided in Table 2 (section I). The time series data and the calculated kinetic parameters are visualized in Appendix A. The effect of oral phenylalanine ingestion [58] (Appendix A) and intravenous phenylalanine infusion [17] (Appendix A) has been examined in healthy individuals. Nuttal et al. [58] found that oral phenylalanine ingestion increased insulin concentrations (iAUC, 3.88 µU/mL/min) compared to water (iAUC, −1.52 µU/mL/min), while glucose concentrations remained unaltered. Co-ingestion of phenylalanine+glucose (Appendix A) resulted in a similar iAUC for insulin compared to glucose ingestion alone, and attenuated the glucose-stimulated glucose response. However, insulin concentrations increased more rapidly and reached higher peak insulin concentrations (64.8 µU/mL) compared to glucose ingestion alone (49.3 µU/mL). Finally, intravenous phenylalanine infusion increased insulin (iAUC, 6.48 µU/mL/min), and decreased glucose concentrations (iAUC, −2.89 mg/dL/min) from baseline in healthy individuals [17]. To our knowledge, no kinetic data are available concerning phenylalanine intake in people with obesity and/or T2DM.

### 3.11. Glutamate

Glutamate study details are provided in Table 2 (section J). The time series data and the calculated kinetic parameters are visualized in Appendix A. The effects of oral glutamate ingestion have been examined in healthy individuals only [59,60,61] (Appendix A).Two out of three studies showed that oral glutamate ingestion increased insulin concentrations (iAUC range, 1.95 to 3.80 µU/mL/min) from baseline [60,61]. Fernstrom et al. [60] showed increased insulin concentrations (iAUC, 1.95 µU/mL/min) after oral glutamate ingestion, compared to the control, whereas the control (a cold flavored vehicle containing 3 g sodium chloride instead of glutamate) did not (iAUC, −0.95 µU/mL/min). Glucose values were not reported. Di Sebastiano [59] showed a non-significant increase in insulin (iAUC, 1.70 µU/mL/min) compared to the control group (gelatin capsules containing NaCl) (iAUC, 0.21 µU/mL/min). Glucose concentrations were unchanged compared to the control group. To our knowledge, no kinetic data are available concerning intravenous glutamate infusion, nor for glutamate intake in people with obesity and/or T2DM patients.

### 3.12. Branched-Chain Amino Acids

BCAA study details (including BCAA composition) are provided in Table 2 (section K). The time series data and the calculated kinetic parameters are visualized in Appendix A. The effect of oral BCAA ingestion (mixtures containing leucine, isoleucine, and valine) has been investigated in healthy individuals only [62,63] (Appendix A). Both studies (excluding the low dose, 1 g, BCAA dose intervention [62]), showed that oral BCAA ingestion increased insulin (iAUC range, 0.47 to 1.51 µU/mL/min), and decreased glucose concentrations (iAUC, −9.22 to −3.67 mg/dL/min) from baseline and the control group (iAUC, −0.29 µU/mL/min, −0.30 mg/dL/min). Furthermore, the 5 g BCAA dose resulted in a higher insulin peak concentration (8.5 µU/mL) than the 1 g BCAA dose (7.3 µU/mL) [62]. The highest insulin response was observed in the study that had the largest BCAA dose [63].

The effect of intravenous BCAA infusion (mixtures containing leucine, isoleucine, and valine) has also been investigated in healthy individuals [63,64,65] (Appendix A). Two out of the three studies showed increased insulin (iAUC range, 0.18 to 0.50 µU/mL/min) [63,64]. Glucose concentrations consistently decreased from baseline in these studies (iAUC, range, −12.05 to −8.37 mg/dL/min). Gojda et al. [63] demonstrated that oral BCAA ingestion increased insulin concentrations (iAUC, 1.51 µU/mL/min) more than the same dose (30.7 g) infused intravenously in healthy individuals (iAUC, 0.42 µU/mL/min). Glucose concentrations declined in the same pattern during both BCAA tests. To our knowledge, no kinetic data are available concerning BCAA intake in people with obesity and/or T2DM.

## 4. Discussion

Amino acids (AAs) have been recognized as important factors involved in glucose homeostasis. In the present systematic review, we aimed to provide a detailed overview of the quantitative effects of oral ingestion and intravenous administration of AAs on postprandial glucose and insulin concentrations in humans. A summary of the results is provided in Table 3.

In total, 55 studies that assessed the effects of 10 AAs, i.e., leucine, isoleucine, alanine, glutamine, arginine, lysine, glycine, proline, phenylalanine, glutamate, and BCAA mixtures were included in this review. The majority of orally ingested AAs, except isoleucine, induced an insulin response when ingested in isolation. Glucose concentrations, with the exception of isoleucine, and lysine, remained unchanged.

The increase in insulin concentrations through AAs has long been recognized, though their mechanisms of action are diverse and not yet fully elucidated [4,69]. It has been shown that AAs affect β-cell insulin secretion through mitochondrial metabolism linked to the tricarboxylic acid (TCA) cycle, and subsequent generation of ATP (e.g., for leucine, glutamine, and alanine) [70,71]. The rise in the ATP/ADP ratio suppresses ATP-sensitive potassium channels, causing depolarization of the β-cell plasma membrane. This in turn activates voltage-gated Ca^2+^ channels and through the influx of Ca^2+^ leads to insulin exocytosis. Other AAs, like arginine via its mCAT2A AA transporter, directly depolarize the β-cell plasma membrane [4,72]. Co-transport of AA with Na^+^ (e.g., alanine, and proline) also depolarizes the β-cell plasma membrane, ultimately leading to Ca^2+^ activated insulin exocytosis [69,73,74,75]. The seemingly unaffected glucose concentrations despite the presence of an insulin response are somewhat surprising. However, compensatory glucagon production and gluconeogenesis to prevent hypoglycemia may at least partly explain these observations [76]. Indeed, multiple studies included in the present systematic review showed increased glucagon concentrations after ingestion of isolated AAs [8,18,19,24,25,26,28,29,31,34,36,37,41,43,45,48,50,53,54,55,58]. Digitizing postprandial glucagon responses found in these studies was, however, outside the scope of the present review. Furthermore, the carbon chain of AA can be used in the liver for gluconeogenesis (i.e., generating glucose from non-carbohydrate carbon substrates), which might further contribute to the lack in suppression of postprandial glucose concentrations [25].

While oral ingestion of most AAs did not alter postprandial glucose responses, oral intake of isoleucine, lysine, and BCAA mixtures evoked a clear decrease in plasma glucose concentrations. Remarkably, the decrease in glucose concentration following isoleucine ingestion occurred without a change in insulin concentrations, suggesting glucose uptake by tissues independent of insulin, as has previously been found in vivo [77]. In the latter study, isoleucine increased glucose uptake in rat skeletal muscle cells through activation of phosphatidylinositol 3-kinase, and independent of mTOR, indicative of insulin-independent glucose uptake.

Co-ingestion of most identified AAs (lysine, glycine, proline, phenylalanine) with glucose did not significantly increase the insulin response compared to the ingestion of glucose alone. However, for these AAs, co-ingestion with glucose prompted a reduction in the glucose response of up to ~70% for phenylalanine. The mechanisms remain to be determined, as it is unclear whether the attenuation of glucose is due to an increased removal rate of glucose (i.e., insulin-independent mechanisms), or due to decreased endogenous glucose production by the liver [55]. Despite the unchanged insulin responses (iAUCs), the postprandial insulin dynamics were frequently altered, indicating a sharper, and more pronounced peak after co-ingestion of AAs (i.e., lysine, proline, and phenylalanine) with glucose as compared to glucose ingesting alone. The greater early rise in insulin concentrations observed after co-ingestion of AAs may imply increased first-phase insulin secretion [54,58], which certainly influences the postprandial glucose concentrations. Co-ingestion of glucose with leucine or isoleucine increased the postprandial insulin concentrations more than the sum of their individual effects, reaching a ~50% increase in iAUC for leucine [8]. This synergistic stimulating effect of the combined intake of AAs and glucose on plasma insulin concentrations was already described by Floyd et al. [6] many years ago. Maximizing insulin secretion could be important in the treatment of T2DM to promote glucose disposal and improve glucose homeostasis [69]. Van Loon et al. [9] showed that co-ingestion of a mixture of protein hydrolysate, leucine, and phenylalanine in long-term T2DM patients, resulted in a considerable (+189%) increase in insulin response compared to the healthy control group (+114%), implying functional β-cells’ secretory capacity to stimuli other than glucose. Manders et al. [78] applied continuous infusion with labeled [6,6-^2^H_2_] glucose to determine blood glucose appearance and disappearance rates following carbohydrate ingestion with or without addition of a protein, leucine, and phenylalanine mixture in T2DM patients. A substantial (~3-fold) greater insulin response was observed following co-ingestion of carbohydrates with AA/protein, with a 28% reduction in blood glucose response, attributed to an increase in plasma glucose disposal.

Intravenous infusion of leucine, arginine, lysine, phenylalanine, and BCAA infusion induced a plasma insulin response and, with the exception of arginine, evoked a decrease in plasma glucose concentrations. These AAs were thus able to induce a substantial increase in insulin response, also observed after oral ingestion, independent from the gut. However, intravenous alanine infusion, unlike oral alanine ingestion, did not induce an insulin response, suggesting that alanine may increase postprandial insulin concentrations through an incretin effect [19]. An incretin effect indicates the release of insulin-inducing substances from the gut and plays a major part in the regulation of postprandial glucose concentrations [79,80]. Incretin hormones, like gastric inhibitory polypeptide (GIP) and glucagon-like peptide-1 (GLP-1), are shown to rapidly stimulate insulin secretion from β-cells in response to nutrients in order to control meal-related glycemic excursions [81]. This incretin effect was also observed by others [82,83], demonstrating that oral ingestion of an AA mixture increased insulin concentrations more than comparable intravenous AA infusion, with increased GIP concentrations. A large number of studies were found that had investigated the effects of intravenous arginine infusion, which is often used to evaluate β-cell function (i.e., during hyperglycemic clamp) and allows for simultaneous examination of acute insulin, c-peptide, and glucagon response [84]. Intravenous arginine infusion stimulated insulin release to a greater extent than oral arginine ingestion; however, studies comparing oral and intravenous administrations are lacking. The larger arginine content in the blood circulation, by avoiding gut metabolism, might have led to increased β-cell stimulation [85,86].

Although limited data are available on the effect of AAs on glucose and insulin responses across different population, studies in obese and T2DM individuals focused on alanine, glutamine, and arginine. Abnormalities in β-cell function are present in prediabetes and T2DM, whereas insulin sensitivity already declines decades before T2DM onset [87]. Furthermore, insulin resistance through excess adiposity is linked to several abnormalities, impacting β-cell function and viability [4]. The studies included here showed that AAs are able to stimulate insulin secretion and lower glucose concentrations in T2DM, and obese individuals. Greenfield et al. [25] observed the greatest insulin response after oral glutamine ingestion in T2DM patients, followed by obese and healthy people. In addition, increased GLP-1 concentrations were found following oral ingestion of glutamine, with no significant differences between T2DM, obese, and healthy individuals. Samocha-Bonet et al. [88] showed that, in patients with well-controlled T2DM, the stimulatory action of GLP-1 on insulin secretion is preserved, reducing postprandial glycemia in T2DM [89]. Whereas oral ingestion of alanine resulted in a comparable insulin response in both healthy and T2DM patients, the insulin dynamics showed considerable differences. More specific, a lower but prolonged elevation in insulin concentrations was observed in T2DM patients [22]. This prolonged elevation in insulin was also observed in obese individuals following intravenous alanine infusion [24]. Intravenous infusion of arginine showed a blunted insulin response in T2DM, as compared to healthy individuals, in three out the four studies. This impairment in insulin secretion might be explained by insufficient β-cell mass, and/or functional defects within the β-cells themselves, in patients with T2DM and individuals at risk for diabetes [4].

In this review, we systematically investigated the effects of AAs on postprandial insulin and glucose dynamics. An extensive approach, consisting of the extraction and utilization of time series data, with a focus on the link between glucose and insulin kinetics, was employed. In the present review, the effects of ten distinct AAs and BCAA mixtures from 55 articles were included. As the AA composition in protein and whole foods largely contributes to the variability observed in glycemic responses, we decided to focus on the glycemic effects of individual AAs. However, the diversity of the included studies, e.g., differences in study set-up, participant characteristics, and measurement instruments, made it difficult to draw quantitative conclusions based on the data. Furthermore, the large heterogeneity in AA dosages used in the studies was not accounted for, when calculating and comparing the postprandial responses, as this would incorrectly assume a linear relationship between AA dosage and postprandial glucose and insulin responses, which we believe is not true. Nevertheless, this might have contributed to a certain extent to the variability observed in postprandial glucose and insulin responses between the different studies. While glucagon measurements were outside the scope of the present systematic review, future studies investigating the effect of AAs on glucose and insulin responses should also include postprandial glucagon concentrations. Notably, since we have made the digitized data on insulin and glucose concentrations publicly available, other parameters such as glucagon concentrations can easily be incorporated in this database. A large difference in the number of identified studies per AA was found in the literature. There was ample information on some AAs (e.g., 28 articles on arginine) and very little information on others (e.g., 1 article on isoleucine), yet several distinct effects were found for the studied AAs. Furthermore, a better understanding of the effects of different AAs on postprandial plasma glucose and insulin responses, as well as putative synergistic effects of co-ingestion of different AAs with glucose, may contribute to the development of more optimal dietary intervention to improve (postprandial) glucose homeostasis.

## Figures and Tables

**Figure 1 nutrients-12-03211-f001:**
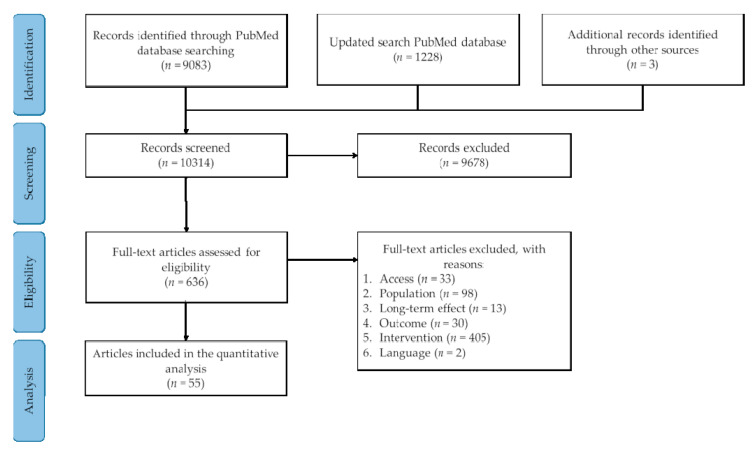
Flow diagram of the systematic literature search.

**Figure 2 nutrients-12-03211-f002:**
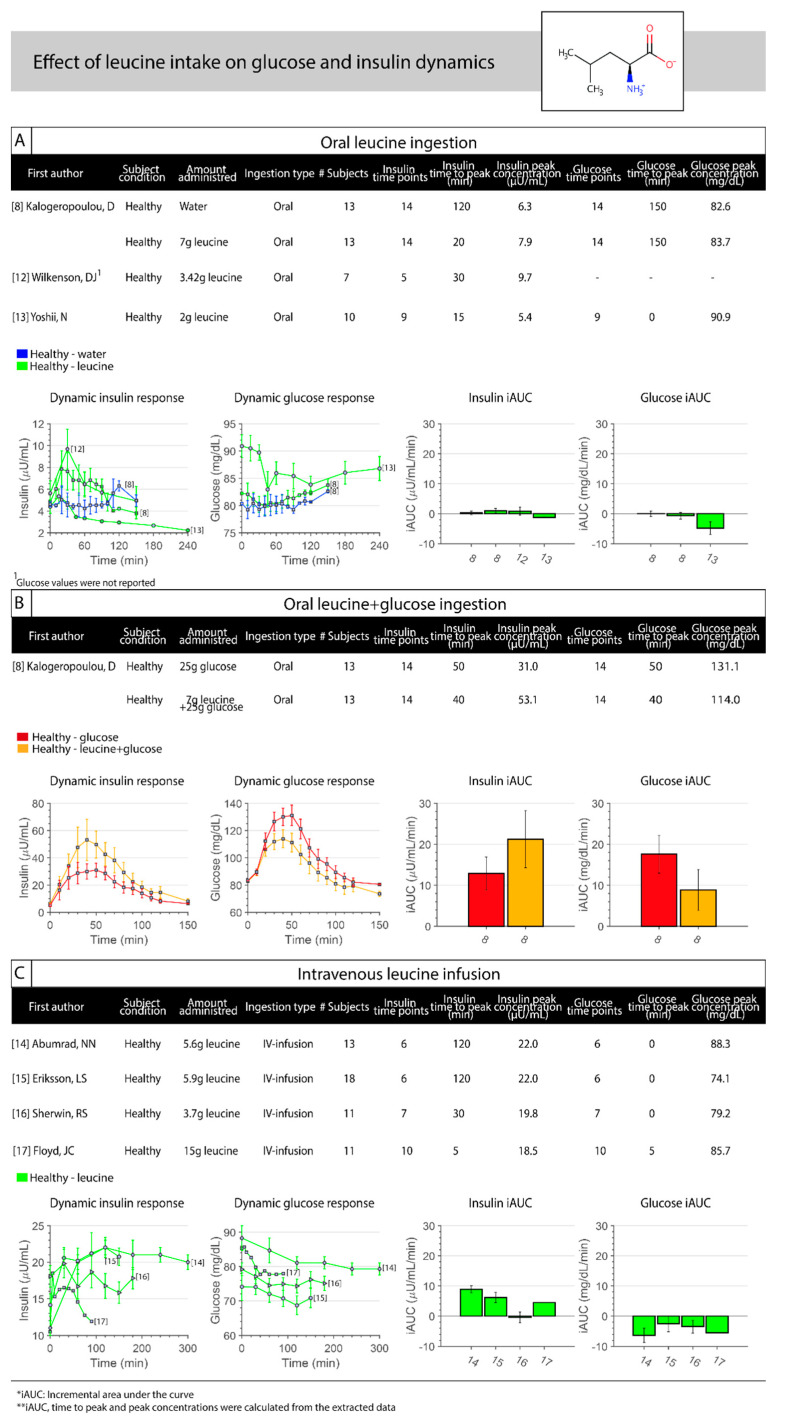
Leucine info-graph consisting of study details with postprandial glucose and insulin time-series data and iAUC after oral leucine ingestion (**A**), leucine co-ingested with glucose (**B**), and intravenous leucine infusion (**C**) in healthy individuals. No data are available for T2DM patients, and obese individuals.

**Table 1 nutrients-12-03211-t001:** Selection criteria.

	Inclusion Criteria	Exclusion Criteria
Population	Healthy adults	Children and adolescents
	Adults with T2DM	Pregnant females
	Adults with overweight/obesity	Animals
		Cells
Intervention	Oral AA ingestion	
	Intravenous AA infusion	
Comparison	Control (i.e., water, saline)	
Outcomes	Glucose concentrations (repeated measurements)	
	Insulin concentrations (repeated measurements)	
Trial design	Intervention study	
Type of publication	Original research articles	Non-English articles
	Published in a peer-reviewed international journal, regardless of publication year	Review articles

T2DM: Type 2 diabetes mellitus, AA: Amino acid.

**Table 2 nutrients-12-03211-t002:** Characteristics of the 55 included studies.

Section	First Author, Year of Publication	*n*	Age (yrs)	BMI (kg/m^2^)	Body Weight (kg)	Lean Body Mass (kg)	Subject Condition	Group	Dose	Route of Administration	Duration (min)	Glucose Measurements	Insulin Measurements	Blood Sampling
	Floyd, 1970 [17]	11	22.9	-	-	-	Healthy	Leucine	15 g	IV	0–30	Blood	Plasma	Forearm vein
		11	22.9	-	-	-	Healthy	Arginine	15 g	IV	0–30	Blood	Plasma	Forearm vein
		11	22.9	-	-	-	Healthy	Lysine	15 g	IV	0–30	Blood	Plasma	Forearm vein
		11	22.9	-	-	-	Healthy	Phenylalanine	15 g	IV	0–30	Blood	Plasma	Forearm vein
**A**	Kalogeropoulou, 2008 [8]	13	24	24	70.9	51	Healthy	Leucine	7 g	Oral	Acute	Serum	Serum	Antecubital or forearm vein
		13	24	24	70.9	51	Healthy	Leucine+Glucose	7 g + 25 g	Oral	Acute	Serum	Serum	Antecubital or forearm vein
		13	24	24	70.9	51	Healthy	Glucose	25 g	Oral	Acute	Serum	Serum	Antecubital or forearm vein
		13	24	24	70.9	51	Healthy	Water	-	Oral	Acute	Serum	Serum	Antecubital or forearm vein
	Wilkenson, 2013 [12]	7	21 ± 0.3	25 ± 0.6	-	-	Healthy	Leucine	3.42 g	Oral	Acute	Plasma	Plasma	Arterialized blood from dorsal capillary bed of the hand
	Yoshii, 2018 [13]	10	25 ± 1	-	65.8 ± 1.5	-	Healthy	Leucine	2 g	Oral	Acute	Blood	Plasma	Cutaneous forearm vein
	Abumrad, 1982 [14]	13	27 ± 1	-	68 ± 2	-	Healthy	Leucine	5.6 g	IV	0–300	Plasma	Plasma	-
	Eriksson, 1983 [15]	18	26 ± 1	22.9	76 ± 2	-	Healthy	Leucine	5.9 g	IV	0–150	Blood	Serum	Arterial blood samples
	Sherwin, 1978 [16]	11	21–35	-	-	-	Healthy	Leucine	3.7 g ^1^	IV	0–180	Plasma	Plasma	Antecubital vein
**B**	Nuttall, 2008 [18]	9	33.8 ± 6.5	28 ± 6.6	81 ± 20.8	56.5 ± 12.3	Healthy	Isoleucine	7.4 g	Oral	Acute	Plasma	Serum	Antecubital vein
		9	33.8 ± 6.5	28 ± 6.6	81 ± 20.8	56.5 ± 12.3	Healthy	Isoleucine+Glucose	7.4 g + 25 g	Oral	Acute	Plasma	Serum	Antecubital vein
		9	33.8 ± 6.5	28 ± 6.6	81 ± 20.8	56.5 ± 12.3	Healthy	Glucose	25 g	Oral	Acute	Plasma	Serum	Antecubital vein
		9	33.8 ± 6.5	28 ± 6.6	81 ± 20.8	56.5 ± 12.3	Healthy	Water	-	Oral	Acute	Plasma	Serum	Antecubital vein
**C**	Rossini, 1975 [19]	6	20–32	-	-	-	Healthy	Alanine	10 g	Oral	Acute	Blood	Serum	Antecubital vein
		6	20–32	-	-	-	Healthy	Alanine	10 g	IV	0–60	Blood	Serum	Antecubital vein
	Rose, 1977 [20]	13	23 ± 4	-	59.5 ± 5.6	-	Healthy	Alanine	11.9 g	Oral	Acute	Plasma	Plasma	Forearm vein
	Sato, 1980 [21]	5	25	-	-	-	Healthy	Alanine	64.6 g ^2^	Oral	Acute	Blood	Plasma	Antecubital vein
	Genuth, 1974 [22]	10	20–46	-	-	-	Healthy	Alanine	33.8 g ^2^	Oral	Acute	Plasma	Plasma	-
		10	20–46	-	-	-	Healthy	Alanine	6.8 g ^2^	Oral	Acute	Plasma	Plasma	-
		10	20–72	-	-	-	T2DM	Alanine	37.5 g ^2^	Oral	Acute	Plasma	Plasma	-
		10	40–71	-	-	-	T2DM	Alanine	8.2 g ^2^	Oral	Acute	Plasma	Plasma	-
	Genuth, 1973 [23]	6	42.5 ± 9.3	-	151 ± 39.6	-	Obese	Alanine	50 g	Oral	Acute	Plasma	Plasma	-
	Asano, 1989 [24]	9	21 ± 0.7	19.8 ± 0.3	53 ± 5	-	Healthy	Alanine	5.3 g	IV	0–2	Plasma	Plasma	Forearm vein
		6	28 ± 3.5	31.3 ± 3	89 ± 12	-	Obese	Alanine	8.9 g	IV	0–2	Plasma	Plasma	Forearm vein
**D**	Greenfield, 2008 [25]	8	30 ± 5.8	21.9 ± 2.2	70.3 ± 8.6	-	Healthy	Glutamine	30 g	Oral	Acute	Plasma	Plasma	Antecubital vein
		8	30 ± 5.8	21.9 ± 2.2	70.3 ± 8.6	-	Healthy	Water	300 mL	Oral	Acute	Plasma	Plasma	Antecubital vein
		8	38.5 ± 8	38.5 ± 6.5	120.6 ± 24.2	-	T2DM	Glutamine	30 g	Oral	Acute	Plasma	Plasma	Antecubital vein
		8	38.5 ± 8	38.5 ± 6.5	120.6 ± 24.2	-	T2DM	Water	300 mL	Oral	Acute	Plasma	Plasma	Antecubital vein
		8	39 ± 9.8	34.5 ± 4.4	106 ± 14.8	-	Obese	Glutamine	30 g	Oral	Acute	Plasma	Plasma	Antecubital vein
		8	39 ± 9.8	34.5 ± 4.4	106 ± 14.8	-	Obese	Water	300 mL	Oral	Acute	Plasma	Plasma	Antecubital vein
**E**	Gannon, 2002 [26]	9	21–52	25.9 ± 0.5	75	61	Healthy	Arginine	10.6 g	Oral	Acute	Plasma	Serum	Forearm vein
		9	21–52	25.9 ± 0.5	75	61	Healthy	Arginine+Glucose	10.6 g + 25 g	Oral	Acute	Plasma	Serum	Forearm vein
		9	21–52	25.9 ± 0.5	75	61	Healthy	Glucose	25 g	Oral	Acute	Plasma	Serum	Forearm vein
		9	21–52	25.9 ± 0.5	75	61	Healthy	Water	-	Oral	Acute	Plasma	Serum	Forearm vein
	Tang, 2013 [27]	8	30.5 ± 3.7	21.0 ± 2.4	56.8 ± 6.8	-	Healthy	Arginine	30 g	Oral	Acute	Serum	Serum	-
		8	30.5 ± 3.7	21.0 ± 2.4	56.8 ± 6.8	-	Healthy	Arginine+Glucose	30 g + 75 g	Oral	Acute	Serum	Serum	-
		8	30.5 ± 3.7	21.0 ± 2.4	56.8 ± 6.8	-	Healthy	Glucose	75 g	Oral	Acute	Serum	Serum	-
		8	30.5 ± 3.7	21.0 ± 2.4	56.8 ± 6.8	-	Healthy	Water	300 mL	Oral	Acute	Serum	Serum	-
	Giugliano, 1997 [28]	10	24 ± 1	23 ± 0.4	67 ± 1.8	-	Healthy	Arginine	30 g	IV	0–30	Plasma	Plasma	Dorsal vein
	Coiro, 1997 [29]	14	24–35	-	-	-	Healthy	Arginine	30 g	IV	0–30	Blood	-	Antecubital vein
	Broglio, 2003 [30]	7	28.3 ± 3.1	29.1 ± 0.9	-	-	Healthy	Arginine	34 g ^3^	IV	0–30	Blood	Blood	Antecubital vein
		7	28.3 ± 3.1	29.1 ± 0.9	-	-	Healthy	Saline	3 mL	IV	Acute	Blood	Blood	Antecubital vein
	Dela, 1990 [31]	7	22	23.2	76	-	Healthy-untrained	Arginine	68.4 g	IV	0–90	Plasma	Plasma	Retrograde direction in a dorsal hand vein (arterialized blood samples)
		7	23	21.6	70	-	Healthy-trained	Arginine	63 g	IV	0–90	Plasma	Plasma	Retrograde direction in a dorsal hand vein (arterialized blood samples)
	Penny, 1970 * [32]	8	-	-	69.5 ± 14.3	-	Healthy	Arginine	34 g	IV	0–30	Plasma	-	Peripheral vein
		8	-	-	69.5 ± 14.3	-	Healthy	Saline	-	IV	0–30	Plasma	-	Peripheral vein
	Levin, 1971 [33]	6	23–50	-	-	-	Healthy	Arginine	20 g	IV	0–20	Blood	-	Antecubital vein
		6	23–50	-	-	-	Healthy	Arginine	20 g	IV	0–20	Blood	-	Antecubital vein
	Bratusch-Marrain, 1979 [34]	9	21–42	-	-	-	Healthy	Arginine	30 g	IV	0–30	Blood	-	Glucose: arterialInsulin: hepatic venous
	Imura, 1976 [35]	15	-	-	-	-	Healthy	Arginine	30 g	IV	0–45	Blood	Plasma	Antecubital vein
	Hasselblatt, 2006 [36]	15	39	24.9	-	-	Healthy	Arginine	30 g	IV	0–30	Blood	Plasma	Glucose: capillary bloodInsulin: Forearm vein
	Berger, 1974 [37]	9	-	-	-	-	Healthy	Arginine	30 g	IV	0–30	Blood	Plasma	-
	Dupre, 1968 [38]	13	-	-	-	-	Healthy	Arginine	15 g	IV	0–40	Blood	Serum	-
		6	-	-	-	-	Healthy	Arginine	3.9 g	IV	0–40	Blood	Serum	-
	Dupre, 1969 [39]	4	18–35	-	-	-	Healthy	Arginine	15 g	IV	0–40	Blood	Serum	Forearm vein
		6	18–35	-	-	-	Healthy	Arginine	15 g	IV	0–40	Blood	Serum	Forearm vein
	Kimber, 2001 [40]	5	58	-	-	-	Healthy	Arginine	31.8 g ^2^	IV	0–30	Plasma	Plasma	Antecubital vein
	Ohneda, 1972 [41]	8	-	-	-	-	Healthy	Arginine	30 g	IV	0–30	Blood	Plasma	-
	Efendic, 1974 [42]	7	30.6 ± 4.4	-	-	-	Healthy	Arginine	27.9 g ^2^	IV	0–30	Blood	Plasma	Brachial vein
		6	36.5 ± 15.3	-	-	-	T2DM	Arginine	29.8 g ^2^	IV	0–30	Blood	Plasma	Brachial vein
	Kawamori, 1980 [43]	9	25–35	-	-	-	Healthy	Arginine	32.3 g ^2^	IV	0–30	Blood	Plasma	Antecubital vein
		5	48–70	-	-	-	T2DM	Arginine	33.5 g ^2^	IV	0–30	Blood	Plasma	Antecubital vein
	Sparks, 1967 [44]	10	25 ± 1	-	66 ± 3	-	Healthy	Arginine	30 g	IV	0–30	Blood	Serum	Venous blood samples
		6	57 ± 3	-	66 ± 9	-	T2DM	Arginine	30 g	IV	0–30	Blood	Serum	Venous blood samples
	Kawamori, 1985 [45]	9	25–30	-	-	-	Healthy	Arginine	32.3 g ^2^	IV	0–30	Blood	Serum	Venous blood samples
		7	48–71	-	-	-	T2DM	Arginine	32.3 g ^2^	IV	0–30	Blood	Plasma	Antecubital vein
	Maccario, 1996 [46]	7	26–32	20.6 ± 1.3	-	-	Healthy	Arginine	27.2 g ^3^	IV	0–30	Plasma	Serum	Antecubital vein
		7	23–52	38.3 ± 2.6	-	-	Obese	Arginine	50.6 g ^3^	IV	0–30	Plasma	Serum	Antecubital vein
	Marfella, 1996 [47]	10	47 ± 0.8	28.1 ± 0.7	78 ± 2.9	-	T2DM	Arginine	30 g	IV	0–30	Plasma	Plasma	Dorsal vein
	Raskin, 1976 [48]	6	48	-	85	-	T2DM	Arginine	20 g	IV	0–40	Plasma	Plasma	Antecubital vein
	Maejima, 2002 [49]	12	58.5 ± 3.3	-	57.0 ± 3.6	-	T2DM	Arginine	30 g	IV	0–30	Blood	-	-
	Ohneda, 1975 [50]	5	63.2 ± 5.9	-	-	-	T2DM	Arginine	30 g	IV	0–30	Blood	Plasma	Glucose: capillary blood Insulin: antecubital vein
	Carpentier, 2001 [51]	14	44.4 ± 2	36.2 ± 1.9	-	-	Obese	Arginine	30 g	IV	−0.75–30	Plasma	Plasma	Forearm distal vein (arterialized venous blood)
	Maccario, 1997 [52]	6	-	-	-	-	Obese	Arginine	49.3 g ^3^	IV	0–30	Plasma	Serum	-
	Walter, 1980 [53]	5	26–40	-	135–227	-	Obese	Arginine	5 g	IV	0–1	Serum	Serum	Antecubital vein
**F**	Kalogeropoulou 2009 [54]	13	30	26	80	60	Healthy	Lysine	11 g	Oral	Acute	Serum	Serum	Antecubital vein
		13	30	26	80	60	Healthy	Lysine+Glucose	11 g + 25 g	Oral	Acute	Serum	Serum	Antecubital vein
		13	30	26	80	60	Healthy	Glucose	25 g	Oral	Acute	Serum	Serum	Antecubital vein
		13	30	26	80	60	Healthy	Water	-	Oral	Acute	Serum	Serum	Antecubital vein
**G**	Gannon, 2002 [55]	9	21–52	25.9 ± 0.5	75	61	Healthy	Glycine	4.6 g	Oral	Acute	Plasma	Serum	Forearm vein
		9	21–52	25.9 ± 0.5	75	61	Healthy	Glycine+Glucose	4.6 g + 25 g	Oral	Acute	Plasma	Serum	Forearm vein
		9	21–52	25.9 ± 0.5	75	61	Healthy	Glucose	25 g	Oral	Acute	Plasma	Serum	Forearm vein
		9	21–52	25.9 ± 0.5	75	61	Healthy	Water	-	Oral	Acute	Plasma	Serum	Forearm vein
	Kasai, 1978 [56]	19	20–70	-	-	-	Healthy	Glycine	22.5 g	Oral	Acute	Blood	Serum	Venous blood samples
**H**	Nuttall, 2004 [57]	8	28	23	80	53	Healthy	Proline	6 g	Oral	Acute	Plasma	Serum	Forearm vein
		8	28	23	80	53	Healthy	Proline+Glucose	6 g + 25 g	Oral	Acute	Plasma	Serum	Forearm vein
		8	28	23	80	53	Healthy	Glucose	25 g	Oral	Acute	Plasma	Serum	Forearm vein
		8	28	23	80	53	Healthy	Water	-	Oral	Acute	Plasma	Serum	Forearm vein
**I**	Nuttall, 2006 [58]	6	26	24	-	59	Healthy	Phenylalanine	9.7 g	Oral	Acute	Plasma	Serum	Antecubital vein
		6	26	24	-	59	Healthy	Phenylalanine+Glucose	9.7 g + 25 g	Oral	Acute	Plasma	Serum	Antecubital vein
		6	26	24	-	59	Healthy	Glucose	25 g	Oral	Acute	Plasma	Serum	Antecubital vein
		6	26	24	-	59	Healthy	Water	-	Oral	Acute	Plasma	Serum	Antecubital vein
**J**	Di Sabastiano, 2013 [59]	9	23.9 ± 1.9	25 ± 2.4	79.4 ± 9	-	Healthy	Glutamate	11.9 g	Oral	Acute	Serum	Serum	Antecubital vein
		9	23.9 ± 1.9	25 ± 2.4	79.4 ± 9	-	Healthy	Gelatin capsules (containing NaCl in the same proportions as in the MSG capsules)	-	Oral	Acute	Serum	Serum	Antecubital vein
	Fernstrom, 1996 [60]	8	25.6 ± 4.1	25.1 ± 1.8	79.7 ± 7.5	-	Healthy	Glutamate	12.7 g	Oral	Acute	Plasma	Plasma	-
		8	25.6 ± 4.1	25.1 ± 1.8	79.7 ± 7.5	-	Healthy	Cold flavored vehicle (containing 3 g sodium chloride instead of MSG)	300 mL	Oral	Acute	Plasma	Plasma	-
	Graham, 2000 [61]	9	26	-	76.9	-	Healthy	Glutamate	11.5 g	Oral	Acute	-	Plasma	Antecubital vein
**K**	Zhang, 2011 [62]	5	22–25	19.7	61 ± 3	-	Healthy	BCAA (weight ratio of 1:2.3:1.2 for isoleucine:leucine:valine)	1 g	Oral	Acute	Plasma	Plasma	-
		5	22–25	19.7	61 ± 3	-	Healthy	BCAA (weight ratio of 1:2.3:1.2 for isoleucine:leucine:valine)	5 g	Oral	Acute	Plasma	Plasma	-
	Gojda, 2017 [63]	18	25.5 ± 1.4	-	76.7 ± 2.6	-	Healthy	BCAA (50% leucine, 25% isoleucine, 25% valine)	30.7 ± 1.1 g	Oral	Acute	Plasma	Serum	-
		18	25.5 ± 1.4	-	76.7 ± 2.6	-	Healthy	Gelatin Capsule (methylcellulose in gelatin capsule, prepared by University Hospital pharmacy)	-	Oral	Acute	Plasma	Serum	-
		18	25.5 ± 1.4	-	76.7 ± 2.6	-	Healthy	BCAA (43% leucine, 24% isoleucine, 33% valine)	30.7 ± 1.1 g	IV	0–120	Plasma	Serum	-
	Tatpati, 2010 [64]	12	23.4 ± 0.8	24.4 ± 1	74.5 ± 4.2	47.6 ± 3.5	Healthy	BCAA (equimolar mixture of valine, leucine and isoleucine)	18.5 g	IV	0–510	Blood	Plasma	Dorsal hand vein (arterialized venous blood samples)
		12	70.7 ± 1.1	24.5 ± 0.7	72.7 ± 3.3	42.2 ± 3.1	Healthy	BCAA (equimolar mixture of valine, leucine and isoleucine)	16.35 g	IV	0–510	Blood	Plasma	Dorsal hand vein (arterialized venous blood samples)
		12	23.4 ± 0.8	24.4 ± 1	74.5 ± 4.2	47.6 ± 3.5	Healthy	Saline	-	IV	0–510	Blood	Plasma	Dorsal hand vein (arterialized venous blood samples)
		12	70.7 ± 1.1	24.5 ± 0.7	72.7 ± 3.3	42.2 ± 3.1	Healthy	Saline	-	IV	0–510	Blood	Plasma	Dorsal hand vein (arterialized venous blood samples)
	Louard, 1990 [65]	10	-	-	-	-	Healthy	BCAA (equimolar mixture of valine, leucine and isoleucine)	3.4 g ^2^	IV	0–180	Blood	Plasma	Arterial samples

^1^ Estimated based on the average body surface area for males and females [66]. ^2^ Estimated based on height/weight tables for males and females [67]. ^3^ Estimated based on average height for males and females [68]. * Two subjects were excluded due to missing time series values. BMI: Body Mass Index, IV: Intravenous, T2DM: Type 2 diabetes mellitus, BCAA: Branched-chain amino acids, *n*: number of participants, -: Data not reported.

**Table 3 nutrients-12-03211-t003:** Summary of the findings.

--	All studies show decrease from control/baseline													
-	One or more studies show decrease from control/baseline													
+/−	No change from control/baseline; contrasting outcomes													
+	One or more studies show increase from control/baseline													
++	All studies show increase from control/baseline													
AA	Oral ingestion						Co-ingestion with glucose		Intravenous infusion					
	Healthy		T2DM		Obese		Healthy		Healthy		T2DM		Obese	
	I	G	I	G	I	G	I	G	I	G	I	G	I	G
Leucine	+	+/−					++	--	+	--				
Isoleucine	+/−	--					++	--						
Alanine	++	-	++	--	++	--			+	+			++	++
Glutamine	++	+/−	++	+/−	++	+/−								
Arginine	+	+/−					+/−	+/−	++	+	++	++	++	++
Lysine	++	--					+/−	--	++	--				
Glycine	+	+/−					+/−	--						
Proline	++	+/−					+/−	--						
Phenylalanine	++	+/−					+/−	--	++	--				
Glutamate	+	+/−												
BCAA mixture (leucine, isoleucine, and valine)	+	--							+	--				

AA: Amino acid, T2DM: Type 2 diabetes mellitus, BCAA: Branched-chain amino acids, G: Glucose concentrations, I: Insulin concentrations. An overview of the studies is provided in Table 2.

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
