# Peer review of "The Impact of Amino Acids on Postprandial Glucose and Insulin Kinetics in Humans: A Quantitative Overview"

_nutrients, 2020, doi:10.3390/nu12103211_

Round 1
Reviewer 1 Report
The review article "The impact of amino acids on postprandial glucose and insulin kinetics in humans: a quantitative overview" presented by Bart van Sloun and coworkers is interesting and useful to the community studying glucose and insulin concentrations in the body and in my opinion should make minor revision.
Authors started with a solid introduction to their literature search. They succeed to justify the importance of having a quantitative review of amino acid effect. The results and the quantitative values are well presented.
There are some minor issues that require attention:
In the abstract: The presented amino acids are all “L” enantiomers. The authors did not write “L-” before amino acids (except glycine of course). L-leucine, L-isoleucine, L-alanine, L-glutamine, L-arginine, L-lysine, glycine, L-proline, L-phenylalanine, L-glutamate…. L-leucine, and L-isoleucine.
The corresponding number “n” near each amino acid can be removed from the abstract. These numbers are mentioned in page 4, line 119-124.
Page 2, line 53, L-AA (“L” is not italic)
Page 21: L-Leucine should be drawn as zwitterions (with NH3+, COO-).
Page 4, line 119-123: Stereochemistry of amino acids might be introduced “L”. It can be added if necessary to all amino acids in the review.
In the supplementary files:
Figures 1 to 10: Amino acids structures should be drawn as zwitterions.
Figure 7: The authors chose to identify the chiral centers when they draw the structures of amino acids, however, they did not identify the chiral center of L-proline.
Author Response
We would like to thank this reviewer for his/her remark that this is an interesting paper and useful to the community studying glucose and insulin concentrations in the body. Furthermore, we appreciate the remark on the well-presented results and quantitative values. We thank the reviewer for the critical comments and useful suggestions to improve the manuscript. We have modified all minor issues suggested by the reviewer.
Minor issues
Q1
In the abstract: The presented amino acids are all “L” enantiomers. The authors did not write “L-” before amino acids (except glycine of course). L-leucine, L-isoleucine, L-alanine, L-glutamine, L-arginine, L-lysine, glycine, L-proline, L-phenylalanine, L-glutamate…. L-leucine, and L-isoleucine.
Answer Q1
We added “L” before amino acids, with the exception of glycine, in the Abstract section of the revised manuscript (page 1, line 25-28).
Q2
The corresponding number “n” near each amino acid can be removed from the abstract. These numbers are mentioned in page 4, line 119-124.
Answer Q2
We removed the corresponding number “n” near the amino acids in the Abstract section of the revised manuscript (page 1, line 25-28).
Q3
Page 2, line 53, L-AA (“L” is not italic)
Answer Q3
We changed the “L” to non-italic in the Introduction section of the revised manuscript (Page 2, line 54).
Q4
Page 21: L-Leucine should be drawn as zwitterions (with NH3+, COO-).
Answer Q4
We updated Figure 2, and included L-leucine as zwitterion in the Result section of the manuscript (Page 21).
Q5
Page 4, line 119-123: Stereochemistry of amino acids might be introduced “L”. It can be added if necessary to all amino acids in the review.
Answer Q5
We added “L” before the amino acids, with the exception of glycine, in the Result section of the revised manuscript (Page 4, line 129-133). In addition, we added a sentence were we explain that any further mention of AAs will refer to their L-form (Page 4, 133-135): ‘For brevity, we will avoid specifying the form of the AAs. Any further mention of AAs will refer to their L-form, with the exception of glycine.
Q6
In the supplementary files: Figures 1 to 10: Amino acids structures should be drawn as zwitterions.
Answer Q6
We updated Figures 1 to 10 in the Supplementary Material, and included the amino acid structures as zwitterions.
Q7
Figure 7: The authors chose to identify the chiral centers when they draw the structures of amino acids, however, they did not identify the chiral center of L-proline.
Answer Q7
We updated Figure 7 in the Supplementary Material, and included the L-proline structure with the chiral center.
Reviewer 2 Report
Here the authors investigated how amino acids affects insulin and glucose secretion based on previously published data.
In general I find the topic interesting and the manuscript well prepared.
I have the following suggestions/questions
1) how you calculated and extracted data was not entirely clear to me, please expand and revise ms
2) you have not considered dose as a potential limitation or explanation in your ms
3) nothing is mentioned about glucagon? but this is highly relevant in the question on insulin and glucose kinetics, see https://pubmed.ncbi.nlm.nih.gov/32255678/
4) the statiscal analyses is not clear to me
5) did you have a hypothesis or a rationale
6) I find the conclusion on that the presented data may aid in t2d a little outside the scope of this manuscript
7) for alanine, I am not sure how the data in your summary table fit with the result section - did I misunderstood?
Author Response
We would like to thank this reviewer for his/her remark that the manuscript is well prepared. We thank the reviewer for the critical comments and useful suggestions to improve the manuscript.
Minor issues
Q1
How you calculated and extracted data was not entirely clear to me, please expand and revise ms Answers Q1
Thank you for this suggestion. We adjusted the Method section of the revised manuscript to include an expanded explanation on how we extracted and calculated data from the identified papers
(page 3, line 103-107): ‘After loading the figures in the digitizing software, the x- and y-axis was set to correctly map the image pixels to the corresponding data values in the figure. Data points (i.e. means and their corresponding standard errors or standard deviations) were determined by manually clicking them and were then stored in Excel data sheets’.
(page 3, line 114-117): ‘The time to glucose and insulin peaks, and their concentrations were identified from the extracted data by selecting the indices with the highest glucose and insulin concentrations, and their corresponding time points in MATLAB.’
Q2
You have not considered dose as a potential limitation or explanation in your ms
Answers Q2
Thank you for addressing this point. The heterogeneity in amino acid dosages provided is large, and might (in part) explain variability observed in postprandial glucose, and insulin responses. While we considered looking into the effect of amino acid dosages, we believe that that the assumption of a linear relationship between dosage and response may be inappropriate, therefore we did not scale the dosages. We expanded upon amino acid dose as a potential limitation in the Discussion section of the revised manuscript (page 28, line 451-454): ‘Furthermore, the large heterogeneity in AA dosages used in the studies was not accounted for, when calculating and comparing the postprandial responses, which might (in part) explain the variability observed in postprandial glucose and insulin responses’.
Q3
Nothing is mentioned about glucagon? but this is highly relevant in the question on insulin and glucose kinetics, see https://pubmed.ncbi.nlm.nih.gov/32255678/
Answers Q3
Thank you for addressing this point. While we already mentioned in the Discussion section that digitizing postprandial glucagon responses was outside the scope of the present review, we agree with the reviewer that glucagon is highly relevant. By making the already digitized data on insulin and glucose publicly available, we have created a platform that can be used by others to add data on other relevant compounds such as glucagon, and c-peptide. We have added this suggestion to the Discussion section of the revised manuscript (page 28, line 454-457): ‘While glucagon measurements were outside the scope of the present systematic review, future studies investigating the effect of AAs on glucagon concentrations are of interest. Furthermore, hormonal responses, like glucagon, can be easily incorporated in the publicly available database we created.’
Q4
The statistical analyses is not clear to me
Answers Q4
Thank you for addressing this point. No statistical analysis was performed in this manuscript due to the heterogeneity of the included studies (e.g. study set-up, participant characteristics, measurement instruments, and amino acid dosages). When talking about significance in the manuscript we refer to the statistics and comparisons made in the original study. We have clarified this in the Method section of the revised manuscript (page 4, line 118-119): ‘The mention of significance in this review refers to the statistics and comparisons made in the original studies’
Q5
Did you have a hypothesis or a rationale
Answers Q5
Thank you for this question. We hypothesize(d) that amino acids exert distinct effects on glucose and insulin dynamics, further influenced by route of administration (i.e. oral, intravenous) and metabolic phenotype (e.g. T2DM, Obese). A better mechanistic understanding would therefore allow for more targeted nutrition studies. We incorporated the hypothesis more explicitly in the Introduction section of the revised manuscript (page 2, line 68-71): ‘We hypothesize that AAs exert distinct effects on glucose and insulin dynamics, further influenced by route of administration and metabolic phenotype. A better mechanistic understanding would therefore allow for more targeted nutrition studies’.
Q6
I find the conclusion on that the presented data may aid in t2d a little outside the scope of this manuscript
Answers Q6
Thank you for addressing this point. We agree that this conclusion is stretching the scope of the manuscript and is too speculative. We therefore changed this sentence in the Discussion section of the revised manuscript (page 28, line 460-462): ‘Furthermore, understanding the synergistic effect of AAs on insulin secretion and the improvement on the glucose regulation may help dietary interventions improve the postprandial glucose response’
.
Q7
For alanine, I am not sure how the data in your summary table fit with the result section - did I misunderstood?
Answers Q7
Thank you for your question. We agree that these results require some clarification. I will elaborate and explain the summary results for alanine based on our scoring criteria (--: All studies show decrease from control/baseline, -: One or more studies show decrease from control/baseline, +/-: No change from control/baseline; contrasting outcomes, +: One or more studies show increase from control/baseline, ++: All studies show increase from control/baseline).
Alanine
Oral ingestion:
-Healthy: All four oral alanine ingestion studies in healthy individuals showed increased insulin (++) from baseline. Oral alanine was found to lower glucose (-) concentrations in one study.
-T2DM: The study on oral alanine ingestion in T2DM patients showed increased insulin (++) and decreased glucose (--) concentrations.
-Obese: The study on oral alanine ingestion in obese individuals showed increased insulin (++) and decreased glucose concentrations (--).
Intravenous infusion
-Healthy: From the two studies on intravenous alanine infusion in healthy individuals, one showed increased insulin (+) and glucose concentrations (+).
-Obese: The study on intravenous alanine infusion in obese individuals showed increased insulin (++) and glucose concentrations (++).
We changed a sentence in the Result section of the revised Manuscript to clarify the effects of intravenous alanine infusion in both healthy individuals and obese participants (page 22, line 198-200): ‘Asano et al. [24] showed increased insulin concentrations in both obese participants (iAUC, 12.02 µU/mL/min) and healthy individuals (iAUC, 3.42 µU/mL/min). Glucose concentrations were increased, despite not being different between groups.